# Comparison of sEMG Onset Detection Methods for Occupational Exoskeletons on Extensive Close-to-Application Data

**DOI:** 10.3390/bioengineering11020119

**Published:** 2024-01-25

**Authors:** Stefan Kreipe, Thomas Helbig, Hartmut Witte, Nikolaus-Peter Schumann, Christoph Anders

**Affiliations:** 1FB Motorik und Pathophysiologie, Klinik für Unfall-, Hand- und Wiederherstellungschirurgie, Universitätsklinikum Jena, 07740 Jena, Germany; 2Fachgebiet Biomechatronik, Institut für Mechatronische Systemintegration, Fakultät für Maschinenbau, Technische Universität Ilmenau, 98693 Ilmenau, Germany

**Keywords:** surface EMG, occupational exoskeletons, onset detection, myoelectric control, constant-false-alarm rate, Teager-Kaiser-Energy-Operator, electromechanical delay

## Abstract

The design of human-machine interfaces of occupational exoskeletons is essential for their successful application, but at the same time demanding. In terms of information gain, biosensoric methods such as surface electromyography (sEMG) can help to achieve intuitive control of the device, for example by reduction of the inherent time latencies of a conventional, non-biosensoric, control scheme. To assess the reliability of sEMG onset detection under close to real-life circumstances, shoulder sEMG of 55 healthy test subjects was recorded during seated free arm lifting movements based on assembly tasks. Known algorithms for sEMG onset detection are reviewed and evaluated regarding application demands. A constant false alarm rate (CFAR) double-threshold detection algorithm was implemented and tested with different features. Feature selection was done by evaluation of signal-to-noise-ratio (SNR), onset sensitivity and precision, as well as timing error and deviation. Results of visual signal inspection by sEMG experts and kinematic signals were used as references. Overall, a CFAR algorithm with Teager-Kaiser-Energy-Operator (TKEO) as feature showed the best results with feature SNR = 14.48 dB, 91% sensitivity, 93% precision. In average, sEMG analysis hinted towards impending movements 215 ms before measurable kinematic changes.

## 1. Introduction

Powered or active exoskeletons are wearable robots designed to assist human movements by enhancing the physical capacity of a person [1,2,3,4]. They are currently developed for applications in medical therapy [4,5,6], support of patients with movement disorders [1,4], as well as in occupational contexts [2,3]. In the latter case, it is hypothesized that occupational exoskeletons can effectively prevent musculoskeletal diseases by assisting the workers in handling excessive loads or by increasing their endurance, thereby limiting muscular fatigue and associated diseases [1,3,7].

For all these applications, a main challenge is the design of a proper human-machine-interface between the powered exoskeleton and its user. Synchronizing the forces generated by the exoskeleton with the muscular efforts of the user is of utmost importance to react properly and timely adequate to the movement intention, and thus provide the required assistance [1,3,4]. This is necessary to ensure biocompatible joint kinematics and dynamics and allow the user to adapt to the system, and gain confidence in its use [3,8,9,10,11].

Surface electromyography (sEMG) is a biosensoric technique, which allows to effectively measure and analyse muscular activity. sEMG signal processing has been investigated for the control of powered exoskeletons [1,4,6,10,12,13,14,15] and in the related field of prosthesis control [11,16,17,18]. Complete decoding of motion intention is not in reach due to the high complexity of the human motor system, its time-varying nature and the high inter-subject variability [4,17]. Still, a broad range of techniques and algorithms has been developed to analyse sEMG with significant success within the corresponding scientific setup. These methods vary strongly in complexity, with the more complex approaches giving better results by accounting for the variability and stochastic nature of sEMG signals [18], but also having higher demands towards computational power and the robustness of measurements [4]. These demands proved as a hindrance to application of more complex sEMG analysis in real life situations: Almost all commercially available exoskeletons for occupational use are passive, and although recent hand prosthesis controllers allow for multiple class pattern recognition, most amputees are only able to reliably use as little as four classes simultaneously.

Considering an active, occupational exoskeleton whose basic control loop is based on inertia and force sensors, follow-up control and limiting of contact forces is possible even without biosensoric information. The disadvantage of these sensor modalities is their inherent time latency, being only able to detect the user motion after its onset. This method, although currently the most common control technique for active exoskeletons, leads to the user necessarily needing to work against the exoskeleton for a short period of time at the beginning of each movement, which is contrary to the goal to reduce muscular effort. Additionally, such an active exoskeleton would not be able to deliver support already at the beginning of the movement, which is important to alleviate stationary weight and friction forces as well as acceleration. sEMG onset detection allows to determine the onset of muscular activity which itself is the origin of all human motion. Due to the electromechanical delay, there is a time span of 20–80 ms between the rise of sEMG and measurable generation of muscular force [19]. This characteristic can be used to eliminate the time latencies of the exoskeleton control loop. In contrast to other sEMG analysis methods, onset detection has low demands towards computational power and robustness of measurements, due to its low complexity [4,20,21].

Many studies on sEMG analysis techniques used simulated data or laboratory investigations far from real-life application cases (i.e., [22,23]). Often, isometric, isokinetic or very simple tasks were used, diminishing the variability of sEMG signals and thereby overestimating the algorithm performance. Also, small groups of test subjects are common. Considering that the variability of the sEMG signal is known to be a main hindrance towards its application, re-evaluation of established methods on a close-to-application dataset of sufficient size is necessary. To this aim, shoulder sEMG of a heterogeneous group of 55 healthy subjects was recorded during seated free arm lifting movements based on assembly tasks. Several state-of-the-art sEMG onset detection techniques are reviewed and compared according to reported performance, computational demands and signal-to-noise-ratio (SNR). A short-list of promising algorithms is implemented and tested on the dataset. The results are compared with kinematic onset detection and onset timings determined through visual inspection of sEMG signals by experts.

The transfer of scientifically developed sEMG signal analysis techniques into real-life applications was rarely achieved up to date. To improve man-machine interaction towards a more synchronous, i.e., physiologically driven behaviour of active exoskeletons, it is of great importance to investigate the established techniques under realistic conditions. In this paper we investigate the performance and reliability of different methods for sEMG onset detection for the use case of an active, occupational shoulder exoskeleton under close-to-application conditions.

## 2. Materials and Methods

### 2.1. Experimental Protocol

The experimental protocol of this study was designed to resemble circumstances close to a real-life application of occupational exoskeletons. Exemplifying seated assembly work, the subjects were asked to perform arm lifts with a low weight (1 kg) while seated in front of a desk. The movements were performed alternating with both arms. The working space was reproduced by using different movement directions (Figure 1) and heights (Placing weight on the table, holding it just above the table, at shoulder height and above head height). All movements have been conducted freely and without any physical restrictions or direct feedback. The subjects were asked to lift the weight with their hand, move it to the target position, stop for at least one second, and return to the starting position, resting there for one second. On each height level, five repetitions of all direction have been performed with each side. Between each height level, a short break was included to avoid muscular fatigue. In total, 55 subjects (29 male, 26 female, 40 “young” (27.2 ± 4.7 years), 15 “old” (52.2 ± 4.2 years), 38 right-handed, 17 left-handed) were included in the study, deliberately chosen to achieve a diverse subject population in terms of sex, age, and handedness. The experimental protocol has been approved by the ethics committee of the Jena University Hospital, Germany (No. 2019-1350_1 BO).

Bipolar sEMG was measured for multiple shoulder muscles using disposable adhesive electrodes (contact diameter 1.6 cm, H93SG, Covidien, Germany). Regarding the electrode placement SENIAM [24] recommendations were followed. Before application of the electrodes the skin was shaved and prepared with abrasive paste. The signal was preconditioned with an analogue 10–700 Hz band pass filter and digitized at 2048 Samples/s and a resolution of 6 nV/bit, using the ToM (Tower of Measurement, Demetec, Germany). Later, the sEMG was conditioned by a digital 20–500 Hz band pass filter and obvious movement were removed semiautomatically. Simultaneously, movement kinematics were recorded using a motion tracking system (Qualisys^®^, Göteborg, Sweden). Both measurements were synchronized using optical trigger signals.

For the sEMG analysis, the data of the anterior deltoids *[M. deltoideus pars anterior]* on both sides was used. Although in a real application of sEMG-based exoskeleton control, probably sEMG of multiple muscles will be included in the decision making, we here focused only on the prime mover muscle for the conducted tasks. To investigate the performance of sEMG onset detection methods, data that reliably contains sEMG activity at movement onsets is needed. Only the anterior deltoid can be expected to provide such muscular activity during the investigated movement. We generally assume that given a proper parameter calibration, any findings achieved for this particular muscle can be transferred to other skeletal muscles. For the kinematic analysis, we utilized the position of the subjects’ hand, indicated by a motion tracking marker.

### 2.2. sEMG Onset Detection Techniques

When muscles are activated, their myoelectric signals were found to show a systematic structure during the phase of initial contraction [11]. Still, according to Drapala et al. [25], as well as others [4,22], precise determination of EMG onset was found to be a challenging task due to the stochastic nature of EMG, smooth gradual transitions from rest to movement, background noise and inter-subject variability. Over the years, several different approaches have been developed to achieve sEMG onset detection.

The task of sEMG onset detection can be split into the steps of signal pre-conditioning, feature extraction, detection and post-processing (Figure 2). Early works addressed the task with the straightforward approach of applying a single or double threshold algorithm, together with a simple time-domain feature, such as mean average value, squared, or rectified and low-pass filtered EMG [26,27,28,29]. Further studies investigated more sophisticated features, such as Teager-Kaiser-Energy-Operator (TKEO) [20,30,31,32,33], sets of optimized time-domain (TD) features [4], sample entropy (SampEn) [23] and wavelet transform (WT)-based time-frequency-domain features [34,35]. In terms of more advanced detectors likelihood-based methods [22,36], Bayesian changepoint analysis [37], Gaussian-mixture-models (GMM) [4,25] and constant false alarm rate (CFAR) adaptive, double thresholding [18,21] have been applied. These methods are summarized in Table 1.

Although processing capacities have increased significantly, computational complexity remains a constraint for online sEMG analysis in real-life applications, especially when considering simultaneous analysis of multiple signal channels such as in the application of an occupational exoskeleton. Therefore, we excluded WT-based features from the comparisons due to their high computational effort [35], even though their performance is among the best reported. Further, Bayesian changepoint analyses, 2-step search algorithms and GMM cannot be used in online applications, because they incorporate the signal after the EMG onset into their decision-making. Likelihood methods are in principle real-time applicable but require training, which is not in the scope of this work.

To assess the performance of sEMG onset detection techniques for real-life applications of occupational exoskeletons, in the following we use a CFAR adaptive, double thresholding algorithm as detector. We compared its performance in combination with different features, namely TKEO, SampEn, the time domain feature set proposed by Trigili et al. [4] (Integrated Absolute Value (IAV), Simple Square Integral (SSI), Waveform Length (WL), Logarithm (LOG)) and variance (VAR) according to Tabie’s and Kirchner’s [20] results. When Trigili et al. [4] evaluated the time domain feature set, they used a GMM with multiple inputs, one for each feature. Since the CFAR algorithm has only one input, we fuse the feature set by taking the average of all four features, each weighted by their mean value to compensate for different numerical dimensions. Root-mean-square amplitude was not investigated separately due to its similarity with VAR. The methods applied are described in detail in Appendix A. The features have been calculated on 50 ms windows with 90% Overlap.

### 2.3. Onset Reference and Evaluation Criteria

When evaluating onset detection algorithm performance, the need for an appropriate reference arises. The gold standard is visual inspection of the sEMG signals by trained experts [21,22,25,35,37]. This method is known to provide the most accurate results, although with some subjective influence of the sEMG examiners. On the other hand, as a manual task, visual inspection requires a huge effort, being unfeasible for larger datasets. The alternative is to reference the detected sEMG onset towards the movement onset which is derived by kinematic analysis [4,20]. From the motion tracking data, we were able to automatically determine the moment, when within the course of one movement the distance covered excesses a certain percentage of the total distance between movement start and end point. We used 5% of within-movement distance of the subject’s hand marker as kinematic onset and as reference point for ground truth for the sEMG analysis. In order to assess the differences to the gold standard method, we visually determined onsets in the sEMG of 4 subjects.

In sEMG onset detection studies, methods are evaluated according to sensitivity, specificity, influence of SNR, and detection latency or timing error [4,18,20,21,22,23,25,34,35]. As in the studies cited above, the algorithm sensitivity is defined as the rate of the movement onsets that are detected by sEMG analysis. We consider a detection as correct, if it is within 500 ms before the kinematic onset reference. In our opinion, specificity is not a feasible criterion here, because in the present experimental context the definition of negative detection is not intuitive. Confronted with the same problem, Tabie and Kirchner [20] used the number of time periods between onsets without detection events as true negatives. Another approach can arise from the fact that it is among the main design goals of an occupational exoskeleton to provide trust to the user. This would be contradicted by every single false positive detection, which in the worst case could initiate a movement not intended by the user. Therefore, we propose the usage of algorithm precision instead of specificity, calculated as 100% minus false positive rate, with the latter being the rate of detected onsets in the sEMG which are not within 500 ms before the kinematic onset reference. In terms of SNR, no additional noise was added artificially (as i.e., in [22]). The SNR was calculated as the feature amplitude ratio in dB between signal areas with and without muscular activation in general and more in detail before movement onset. Therefore, the visually inspected and marked data was used. Finally, as already stated by Trigili et al. [4], in control applications the time delay between detected sEMG onset and movement onset is not a bug, but a feature. Where in clinical applications, the aim of automated sEMG onset detection is to replace manual, visual inspection and therefore be as close to the manual detection as possible, this is not relevant for the application in the control of powered exoskeletons. There, the algorithm is only required to detect muscular activity early enough before the start of a movement to implement exoskeleton pre-control. No fixed minimum time difference can be given a priori, but rather has the exoskeleton control to be fast enough to react in the remaining time. We therefore rate higher onset delays as better, as long as they are within a reasonable range of up to 500 ms before movement onset. However, as long as the onset detection exhibits a sufficient time lag towards the kinematic reference the exact timing of an algorithm is secondary to its sensitivity and precision.

## 3. Results

For the example of one movement onset, the sEMG as well as the feature curves are shown in Figure 3. There, the area visually found to contain muscular activation is highlighted in grey. The dotted line with the ‘5%’ mark shows the kinematic onset reference timestamp.

The SNR of all features was calculated as their amplitude ratio in dB between signal areas manually labelled as with muscular activity and without. Additionally, contrast values in windows 300 ms before and after the onset of muscular activity were taken into consideration. The results are displayed in Table 2. It is obvious that TKEO shows the highest SNR values in all categories. The variance feature shows slightly increased values compared to the mean of the time-domain features proposed by Trigili [4]. In contrast, SampEn shows very low SNR values, but more consistently, reaching its maximum value almost directly after onset.

The combinations of the CFAR adaptive, double threshold and each feature were implemented and tested separately. To ensure optimal detector performance for each feature, the thresholding algorithms’ parameters were optimized for each feature to reach a maximum mean value of Sensitivity and Precision (MSP) on a data subset.

All movement onsets of all subjects were included into the evaluation (total: approx. 11,000 onsets). The accuracy of each feature-detector combination is displayed in Figure 4. As comparison, the accuracy of the visual inspection is shown for both experts (MAN1, MAN2). The TKEO + CFAR method shows the highest accuracy values of all tested algorithms. Its sensitivity and precision is about 4% lower than both visual detections.

The onset detection latencies for each method are shown in histogram form in Figure 5. It needs to be remembered that different data sizes amounts are displayed for the time differences of each method. The visual inspection covered only approx. 900 onsets due to its limitation on 4 subjects. In contrast, the algorithms have been tested on approx. 11,000 onsets, but since time differences could only be calculated for successful detections, the amount of onsets taken into account vary between the algorithms by their sensitivity.

Most visually detected onsets showed a time difference of more than 150 ms to the kinematic reference point. The automatically detected onsets exhibited a larger variability and less time difference with most detections before 100 ms towards the kinematic reference. The onset time differences of all methods were statistically tested using a mixed linear model. Although the onset time differences of the methods show significant differences to each other, but with only small to medium effect sizes.

## 4. Discussion

Due to the electromechanical processes during muscle activation, there are time delays between EMG onset, muscle force generation, and movement onset. The delay between EMG and force generation was named electromechanical delay (EMD) by Cavanagh and Komi [38] and is in the order of 20–80 ms [19]. The movement onset (kinematic reference) might be delayed even further from the EMG onset because of inertia and forces opposing the movement, up to several hundred milliseconds [4,20,35]. This is represented by the results in Figure 5, showing an average time delay of about 200 ms to 5% of the distance covered during the movement. Since the manual inspection can be assumed to be as accurate as possible, its time difference towards the kinematic reference is close to the real time difference between the onset of muscular activation and the movement onset. Therefore, regarded inversely, the histograms of the manual, visual inspection represent the spread in time delay introduced by the kinematic reference method. Factors contributing to this variability may be other muscles initiating the movement or muscular co-contractions overlaying the onset. Compared to similar studies [4,20,21,37], the time differences here are found to be even slightly higher, possibly due to the use of an additional weight in the experimental protocol. Concerning the algorithms for sEMG onset detection, the onset time differences and their deviation are in the expected range. Their statistical indifference means that all methods under investigation exhibit no relevant performance differences regarding the onset detection timing error.

The visual inspection method reaches only 95% sensitivity and 97% precision although commonly perceived to detect all muscular activity present in the sEMG signal. This finding is probable to be influenced by the same causes as the variability of the onset time differences and shows the difficulty of non-laboratory sEMG onset detection, influenced by temporally low muscle activation, combinations of multiple muscles contributing to a certain movement, and the presence of noise in the signal. This becomes clear when reviewing the high sensitivity values reported by other studies with simulated data or signals measured under laboratory conditions [18,22,23]. Therefore, 100% onset detection performance is not to be expected when regarding a single muscle, even if it is the prime mover such as the *M. deltoideus pars clavicularis* for the movements investigated here.

Regarding the automated onset detection methods under investigation, the TKEO + CFAR combination performed best at sEMG onset detection under close-to-application. Its sensitivity and precision were 91% and 93% respectively, only 4% less than the average of visual inspection results by sEMG experts. The significance of algorithm sensitivity and precision becomes especially clear, when regarding the error rates: 91% sensitivity means 9% of movement onsets not being detected, while 83% sensitivity means 17% of movement onsets not being detected, which is almost the double. In the same perspective, 93% precision means 7% of the detected onsets not being in relation with an upcoming movement, resulting in possible unwanted actions of the exoskeleton. This highlights the value of the higher performance of the TKEO + CFAR technique versus the other methods. However, with continuous detection regular false alarms are still to be expected. In applications of occupational exoskeletons, this issue needs to be dealt with by post-processing, considering multiple muscles or a robust control scheme in order to avoid unintended movements and to ensure trust by the user in the assistive device.

## 5. Conclusions

The aim of this study was to investigate the performance of sEMG onset detection methods under close-to-application conditions of active exoskeletons. Therefore, free and unconstrained movements representing the workspace during seated manual assembly tasks have been investigated on a large and heterogeneous group of subjects. This unique dataset with approx. 11,000 movement onsets allowed testing of known and promising sEMG onset detection algorithms regarding their performance in a real application case unlike the common, limited laboratory setting.

For a reduced dataset, the gold standard for onset detection, manual, visual inspection allowed to evaluate the possible onset detection sensitivity and precision when only regarding the prime mover muscle, showing that before 95% of movement onsets sEMG activity can be found and that 97% of sEMG onsets happen shortly before a movement onset. It thereby also permitted evaluation of kinematic reference as an automated method to generate a ground truth for onset detection in larger datasets and possibly even during online testing.

The computationally simple approach with TKEO + CFAR showed good results for sEMG onset detection under close-to-application condition (91% sensitivity, 93% precision), performing best within the set of applied methods. Still, reliable sEMG onset detection in applications remains a challenging task, with accuracies not reaching 100% even if manual visual inspection, still serving as the gold standard is applied. Anyhow, time-frequency-features based algorithms and data of multiple muscles bear the potential to improve the presented results, but at the cost of necessarily higher computing cost. No matter the actual technique used in a real application, the results presented here show that the sEMG contains sufficient information for timely pre-detection of movement onsets and therefore exoskeleton pre-control even under close-to-application conditions.

## Figures and Tables

**Figure 1 bioengineering-11-00119-f001:**
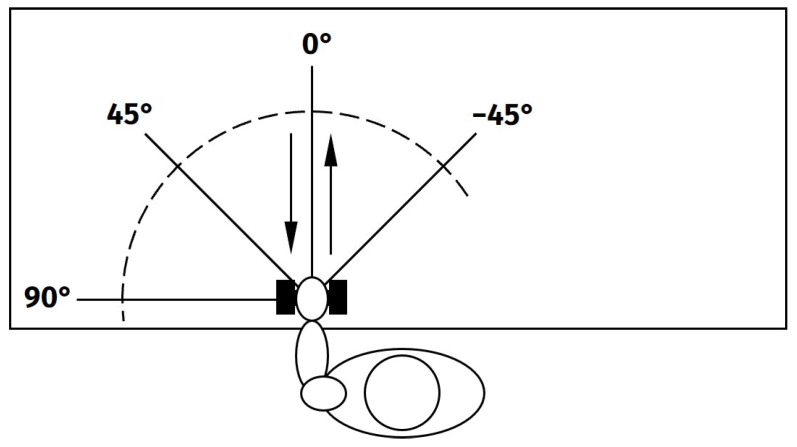
Movement directions, projected onto table plane. Movements for left arm are shown, but movement were done with both arms.

**Figure 2 bioengineering-11-00119-f002:**
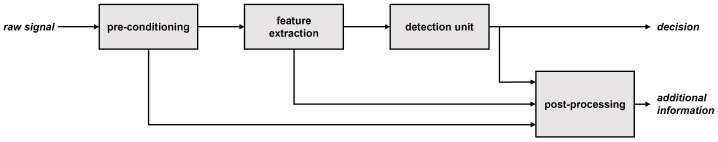
sEMG onset detection processing structure.

**Figure 3 bioengineering-11-00119-f003:**
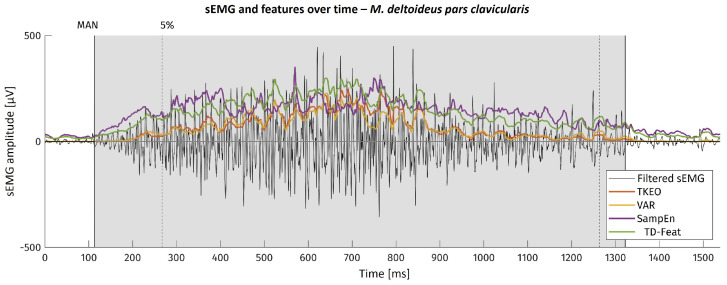
sEMG at movement onset and feature curves, exemplarily shown for one signal section. Area marked in grey: Visual detection of muscular activity. Dotted line with ‘5%’ mark: Timestamp of kinematic onset reference.

**Figure 4 bioengineering-11-00119-f004:**
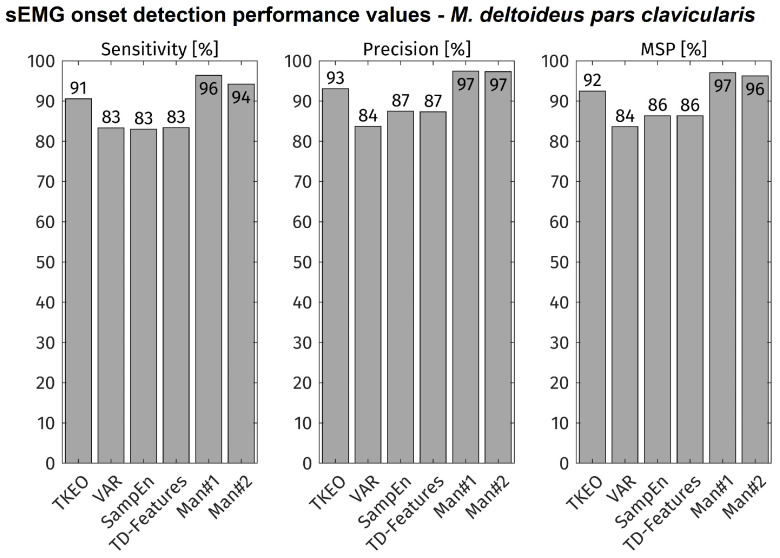
sEMG onset detection performance values. Sensitivity is true positive rate, precision is 100% minus false positive rate and MSP is the mean of sensitivity and precision. ‘Man#1’ and ‘Man#2’ depict the results of the manual, visual inspection by experts. Manual inspection: ca. 900 movement onsets under investigation; Automatic detection: ca. 11,000 movement onsets under investigation.

**Figure 5 bioengineering-11-00119-f005:**
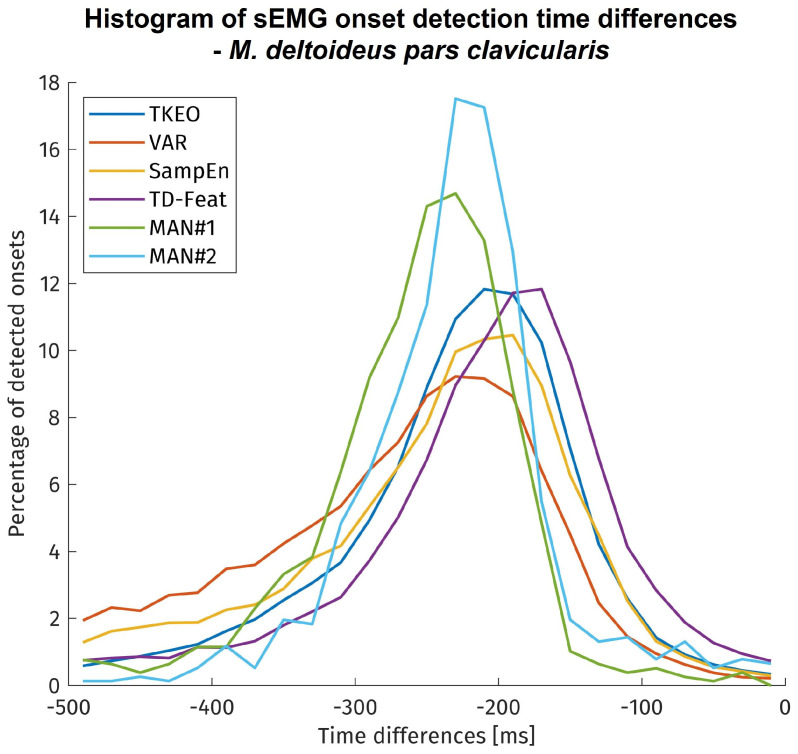
Histogram of the sEMG onset detection time differences to the ‘5%’ kinematic reference. ‘MAN#1’ and ‘MAN#2’ depict the results of the manual, visual inspection by experts. Manual inspection: ca. 900 movement onsets under investigation; Automatic detection: ca. 11000 movement onsets under investigation.

**Table 1 bioengineering-11-00119-t001:** sEMG onset detection methods.

Type	Method	Reference(s)
Features	TKEO	[20,30,31,32,33]
	optimized TD feature set	[4]
	SampEn	[23]
	WT features	[34,35]
Detectors	likelihood-based methods	[22,36]
	Bayesian changepoint analysis	[37]
	GMM	[4,25]
	CFAR adaptive tresholding	[18,21]

**Table 2 bioengineering-11-00119-t002:** sEMG onset detection feature SNR in decibel.

	TKEO	VAR	SampEn	TD-Features
**Total SNR**	14.84 dB	13.64 dB	4.44 dB	13.42 dB
**Onset SNR**	10.59 dB	10.29 dB	4.15 dB	10.10 dB

## Data Availability

The data presented in this study are available on request from the corresponding author.

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
