# Peer review of "Comparison of sEMG Onset Detection Methods for Occupational Exoskeletons on Extensive Close-to-Application Data"

_bioengineering, 2024, doi:10.3390/bioengineering11020119_

Round 1
Reviewer 1 Report
Comments and Suggestions for Authors
This study evaluated the sEMG onset detection performance using different features with the CFAR detection algorithm. The results demonstrated that the CFAR with the TKEO feature showed the best performance. The aim of the study is clear, and the manuscript is well-organized, while several limitations prohibit the reviewer from seeing the significance.
1. The main limitation is that this study only evaluates the performance of previously proposed methods in a new situation (shoulder muscles). Also, the methods tested in this manuscript have similar performance. Therefore, the conclusion is weak, and the potential applications are limited.
2. Specific comments
a) The Introduction section does not describe the state-of-the-art onset detection methods, their performance, and application situations.
b) A brief introduction of the movement onset method by kinematic analysis is suggested to be provided.
c) What’s the definition of sensitivity and precision in the onset detection?
d) The feature curves around the onset should be provided to illustrate the difference between features better.
e) Line 176, ‘The relative timing of the kinematic onset is shown for 1% and 5% of the distance covered during the movement.’ Is difficult to understand. What’s the meaning of 1% and 5% distance? So as the Figure 3. Why is the time difference between the visual onset and kinematic reference so large (>100 ms)?
f) Is there any difference in the performance across different tasks? 0, 45, and 90 degrees?
g) In figure 5, the onset detection performance for different features and visual detection is similar. Although the authors concluded that TKEO + CFAR showed the best results, will these slight differences significantly influence the performance in practical applications?
h) The Discussion is simple and superficial. The possible reasons accounting for the performance difference between features should be discussed. In addition, a comparison with previous studies and potential applications of this study should be provided.
Reviewer 2 Report
Comments and Suggestions for Authors
General topics
The paper is very useful because it reviews different techniques for accurately defining the onset of muscle activity through the analysis of surface EMG. In general, the paper is well written and clear. The comparative study would help to unify different proposals to solve such an important issue when the final goal is to set the feedback system in an active exoskeleton.
However, the innovative contribution of the work is not clear. It is true that it is very interesting to carry out a comparative study between different analysis techniques, but it should provide its own research, some technique, some important modification to improve the results, some new recording protocol, etc. Other authors have already carried out the proposed analyses, and the experimental protocol is simple.
The title of the paper is very general, when the defined objective is just to compare activation detection techniques, as quoted in lines 63-64.
Only one very specific muscle (deltoid) has been recorded, when in exoskeleton feasibility reports there are more and more studies of muscle relationships. This multidimensional aspect should be discussed in the paper; in fact, the authors also briefly express the limitation of the study (line 225).
Introduction
Line 12. Is the SNR value in decibels? It should be, although we will see later that it does not refer to a signal parameter, but rather to the ratio of the calculated parameters between rest and exercise. The concept of SNR needs to be better clarified at the beginning.
The introduction does not include a last sentence as a conclusion, revealing the importance of the machine being ahead of the action. In this sense, in the introduction, it is very important to say whether the study is for active or passive exoskeletons; knowing the activation times is not so important for passive exoskeletons.
Material and Methods
Line 83. It is not clear from the experimental protocol whether the arm performing the action is the dominant arm or not. Figure 1 seems to indicate that it is the non-dominant arm. Furthermore, in this first paragraph of the experimental protocol it is not clear at what height or heights the subjects lift the weight.
A graph of the recording of the sEMG signal should be added, which can be used to explain the experimental protocol more clearly, but also to know the type of recordings to be analyzed. When the recordings are dynamic, this aspect is very delicate, especially when the high-pass filter of the signal is set at 10 Hz (possible movement artefacts). The timing of the exercise can be shown on this figure.
Line 94. The dimensions of the electrode contact must be given.
Line 101-102. Electrode placement should be given, or if the SENIAM protocol has been followed, referenced.
Line 111. Can be left as "sEMG", in your abbreviations table it appears like this.
Line 119. Enter the abbreviation as it appears in Table 1: "..optimized time-domain (TD) features..".
Lines 135-141. It seems that the simple techniques based on the root mean square value of the signal that the authors indicate (lines 114-117) have also been discarded. They should explain why they have discarded this type of feature, which in the end usually gives good results, or whether they have included it in the group of "optimized time-domain features".
Lines 137-141. In fact, only one detector is used, and four features. This should be clarified in this paragraph. It is true that Appendix A explains the mathematics of the parameters, but at the end, they calculate the average of four of them (SSI, IAV, LOG, WL). This is one of the aspects that causes most doubts in the article; they should explain why they take the average of four parameters (which are dimensionally different parameters); in addition, the authors analyze separately (individually) the most classical one, which is the variance of the signal (or the root mean square with moving window). This must be clarified.
Line 164. It is clear that the gold standard is visual inspection, but that it is very costly in terms of time and effort when there are many records. For this reason, an automatic analysis of the capture of movements is carried out, in order to redefine it as the gold standard with the results to be analysed. But as said before (line 169) the difference between what the experts see in the sEMG and what the movement analysis detects is not an error, it is precisely what we want to determine in this study. I think this paragraph should clarify this aspect, and make it clear what the results are to be compared with, because the visual analysis of all sEMG records has not been done.
Results
Figure 3. How many activations does it represent? It seems to be the exercises of four subjects, as stated (line 173), but how many movements are there in total? It must be said here to understand the figure, even if it is repeated later (line 201). In addition, how were these four subjects selected?
Table 2. Looking at the results in table 2, SNR values, it seems that this is a new definition of SNR, which traditionally is the ratio of the signal in a muscle activation section and a resting section, calculated in dB. It should be explained how SNR has been calculated because it seems to refer to a ratio between the features studied, not a ratio of the signal.
Line 205. It is not "well above ….", it is just above.
Line 207. The comments are qualitative, and a statistical study should be carried out, although it is unlikely to provide significant differences between any of the 6 techniques. If there are no significant differences, it is very difficult to defend one technique over another.
Throughout the text, medians are compared, when the distributions of the data obtained are likely to follow the normal distribution (has the normality study been carried out?), in which case the means can be compared.
In the introduction (lines 13-14) a result is reported that does not appear in the results chapter. It must be explained in Results. TKEO is quoted as having a timing error of -25 ms. But we also know that there is a time lag between myoelectric activation and cell activation, so it is not really an error, is it? In addition, what values do the other techniques show for this supposed error?
Discussion
Line 228. TKEO + CFAR seems to achieve better onset detection, but this is not a significant improvement over the other techniques, or at least no conclusive results have been presented in this regard.
Figure 5. Figure should be reinforced with a table showing the significant values of means (or medians) and standard deviations. If this is not done, the work seems more evaluative than rigorous, and the discussion cannot be conclusive.
The overall discussion is very brief, and there are some questions to be answered. The technique defended as "the best" is made on the basis of sensitivity and accuracy results, but do we know if it is logical that it is the one that gives the least difference with respect to movement onset? It also appears that there are many values, for the TKEO technique, outside the statistical or whisker range (figure 5). Why have spectral parameters not been considered in the study? The TD-features technique is actually a mixture of conceptually distinct parameters, what would have happened if each of them were worked with separately?
Conclusions
Line 247. With the results provided and without comparative statistics, it cannot be said to be the best technique.
Line 250. Time-frequency techniques should certainly have been included in the study. There is no need to study wavelets, a simple spectrogram would also have returned interesting results. Why has this study not been carried out?
References
References that are theses or master's theses should be substituted, because if they were relevant, they surely led to indexed publications.
Round 2
Reviewer 1 Report
Comments and Suggestions for Authors
All the comments suggested in the previous reviews are fully attended and the manuscript is now ready for publishing.
Author Response
Thank you fo the positive evaluation of our revison.
Reviewer 2 Report
Comments and Suggestions for Authors
Line 126, the sentence is unfinished.
The statistical study of the time differences (figure 5) must be carried out. Regardless of the fact that the values of sensitivity and precision are important, accurately locating the onset of muscle activation is the ultimate goal of the study. If the curves represented in figure 5 do not show significant differences, for example at line 298, it will mean that any of the methods is valid for locating this onset time.
Reference 10 is still a PhD thesis
